# Overcoming Barriers in the Introduction of Early Warning Scores for Prevention of In-Hospital Cardiac Arrests in Austrian Medical Centers [note 1]

**DOI:** 10.3390/healthcare13202624

**Published:** 2025-10-18

**Authors:** Benedikt Treml, Philipp Dahlmann, Sasa Rajsic, Lydia Bauernfeind

**Affiliations:** 1Department of Anesthesia and Intensive Care Medicine, Medical University of Innsbruck, Health University of Applied Sciences Tyrol, 6020 Innsbruck, Austria; benedikt.treml@tirol-kliniken.at; 2Department of Anesthesiology and Intensive Care Medicine, Medical University of Innsbruck, 6020 Innsbruck, Austria; 3Faculty of Applied Healthcare Sciences, Deggendorf Institute of Technology, 94469 Deggendorf, Germany; philipp.dahlmann@th-deg.de

**Keywords:** in-hospital cardiac arrest, survey, adults, early warning scores, shortage of resources, process management, change management, best practice tutorial

## Abstract

**Introduction**: In-hospital cardiac arrest (IHCA) is still associated with high mortality. Introduction of multi-parameter early warning systems (EWS) could reduce the incidence of IHCA. However, data regarding prevention of IHCA remains conflicting. Moreover, an aging population and a shortage of healthcare workers strain Austrian acute care hospitals. Sicker patients and fewer staff could hinder the implementation of multi-parameter EWS in Austria. Therefore, we sought to identify such barriers by assessing local and national data. Furthermore, we investigated the incidence of in-hospital cardiac arrests at Medical University Innsbruck. **Methods**: In this perspective study, we retrospectively analyzed all patients experiencing an in-hospital cardiac arrest between 2017 and 2024. In the qualitative part, ten experts in in-hospital emergency medicine were interviewed using guided interviews. The main results from the interviews were identified using a structured content analysis according to Mayring. Quantitative and qualitative data were integrated through narrative. Using the Consolidated Framework for Implementation Research, we stratified our data into five domains. Finally, we applied the “eight steps for leading change” to develop a practice guideline. **Results**: In six years, 1356 patients were treated by an emergency medical team; 1317 emergencies were included, with 365 of them being resuscitated. Overall, 114 survived for 24 h. The incidence rate of in-hospital cardiac arrests was 0.86 cases/1000 admissions per year. The guided interviews demonstrated a nearly complete absence of EWS using multiple parameters in Austria. Strained human resources after the pandemic, the fear of an increased workload and the lack of robust data regarding the benefit of survival were mentioned as main reasons. The best practice tutorial considers the challenges identified and provides guidance for structured implementation in hospitals. **Conclusions**: Implementing NEWS2 can facilitate detection of critically ill patients despite decreased staffing. Identifying common barriers and facilitators in five domains described and applying this to the “eight steps for leading change” enabled us to provide a tutorial for implementation of an EWS. This could help master future challenges in in-hospital emergency medicine.

## 1. Introduction

In Austria, a dense network of acute care hospitals exists, with approximately seven hospital beds per 1000 inhabitants [1]. Twenty-eight specialized hospitals provide an extended range of medical services beyond basic care. Seven central hospitals offer comprehensive care 24/7 for acute and complex medical needs. These hospitals treat a plethora of patients and in-hospital emergencies. One of the worst emergencies, in-hospital cardiac arrest (IHCA), is still associated with high mortality. Given the incidence of IHCA ranging from 1.5 to 2.8 per 1000 in-patient hospital admissions in Europe [2], emergency medical (critical care outreach, CCO) teams of such centers face the challenge of a timely response. One of several attempts to improve survival in IHCA was the implementation of multi-parameter early warning systems (EWS). It has been shown that 80% of patients on general wards exhibit significant clinical deterioration within 24 h prior to an IHCA [3,4,5,6]. Therefore, clinical decision-support tools were designed to detect early signs of patient deterioration. In short, hospitalized patients undergo routine vital sign measurements to detect clinical deterioration (“the tracking”). This is combined with the calculation of the EWS score: the worse the patient’s condition is, the higher the score sums up. When reaching predefined thresholds, alerts are triggered to prompt clinicians to take predefined actions based on score levels. This is often referred to as “track and trigger” systems. Other EWS trigger alerts are based on single-parameter thresholds, such as a respiratory rate > 30 or systolic BP < 90 mmHg. In summary, EWS aims to facilitate the timely recognition of patient deterioration outside critical care settings and ensure prompt intervention by appropriately skilled personnel. Furthermore, this early detection could prevent futile critical care as well as permanent damage to the health of our patients.

Since 2000, a plethora of EWSs have been introduced; one of the best evaluated EWSs is the second version of the NEWS assessment tool [7]. In the validation study of the NEWS, it was shown that this score was better at predicting unplanned ICU admission and death within 24 h than 33 other early warning systems, but not the occurrence of IHCA [8,9]. In England, the NEWS had been implemented in about three-quarters of acute hospitals by 2015. This measure reduced the incidence of IHCA by almost 10% [10]. In the first version of NEWS, a normal value of SpO_2_ of ≥96% and an SpO_2_ of ≤91% already triggered the maximum point value of 3 in this category, so excessive use of oxygen could occur in chronic lung diseases [11]. This was improved in 2017 with an update to a newer second version (NEWS2) [7]. However, this increased the complexity of the score and reduced its sensitivity [12]. Furthermore, inconsistent success in reducing mortality through the implementation of EWS has been shown [13,14]. This may be due to methodological weaknesses in the validation of many EWSs [13]. Moreover, good performance in some subgroups does not warrant its use in all diseases and all clinical settings [15].

Expeditious detection of critical patients outside of highly specialized intensive care units still poses difficulties as personnel resources are getting strained more and more in Europe [16]. At the same time, an aging Austrian population will lead to sicker and older patients in healthcare [17].

Looking at the processes involved in in-hospital emergencies enables efficient use of already scarce resources [18]. A wide variety of professional groups and specialist areas interact in healthcare and face increasing demands on effectiveness and quality. Moreover, the amount of time that healthcare professionals spend directly with the patient is decreasing. This is even more relevant as the assessment of EWS takes more time than simple measurement of vital signs even if being facilitated electronically [19,20]. This may be a barrier from the perspective of the staff on the wards. Additionally, after the pandemic, interest in the use of wearable devices for ease of continuous monitoring of vital signs grew [21].

Despite this shortage of time in healthcare, recent ERC recommendations for the usage of EWS disregard substantial differences between different healthcare systems [22]. This leads to a distinct gap in research: do barriers exist when, e.g., NEWS2, developed in the UK, is transferred to any other country without modifications?

Our primary objective was to provide a contiguous overview of the local situation regarding in-hospital cardiac arrests at Medical University Innsbruck. Therefore, we first investigated the rate of IHCA and its survival at the Medical University Innsbruck.

Based on this data, we sought to consult experts in adult in-hospital emergency medicine from central and specialized hospitals in Austria. Semi-structured interviews should investigate the current situation of adult in-hospital emergency medicine and the distribution of EWSs in Austria. Moreover, we aimed to identify barriers during the implementation of an EWS in such hospitals.

Lastly, we applied the Consolidated Framework for Implementation Research (CFIR) to stratify our local data into five domains when implementing an EWS. By describing the domains of innovation, of the outer setting, of the inner setting domain, of the individual, and of the implementation of the process, respectively, we sought to structure the available literature and local data from different perspectives. Finally, we wanted to provide guidance for implementation in a real-world setting by using Kotter’s “eight steps for leading change” [23].

## 2. Materials and Methods

### 2.1. Study Design

We analyzed quantitative data regarding the local incidence of IHCA first. Subsequently, we conducted a qualitative analysis for an in-depth understanding of current in-hospital emergency medicine in Austria. Through a cohesive report, the quantitative and qualitative data strands were weaved together. Using the Consolidated Framework for Implementation Research (CFIR) allowed us to explain barriers and facilitators for the implementation effectiveness of EWS. Finally, we created a best-practice tutorial according to the “eight steps for leading change” [22].

### 2.2. Quantitative Methods

#### 2.2.1. Subjects

We included all patients experiencing an in-hospital cardiac emergency between 1 January 2017 and 31 December 2022 at the Medical University Innsbruck. Patients aged < 18 years were excluded.

Depending on the location of an IHCA, one of two adult CCO teams is alerted 24/7 using structured alarm criteria (single track-and-trigger system) as recommended by the ERC guidelines [22]. They comprise a minimum of an intensivist and two intensive care nurses. All patients received post-resuscitation care at one of the tertiary intensive care units of the Department of Anaesthesiology and Critical Care Medicine and the Department of Internal Medicine, Medical University Innsbruck, Austria. These ICUs treat surgical, post-trauma, and medical patients.

#### 2.2.2. Collection of Quantative Data

We prepared our retrospective work according to “The strengthening the reporting of the observational studies in epidemiology (STROBE)” statement [24] (Appendix A). We reviewed the electronic records of both adult hospital CCO teams. We identified the number of all in-hospital emergency calls and all patients who were treated by a CCO team. Thereafter, we collected the rate of IHCA and the rate of return of spontaneous circulation (ROSC) from the annual in-hospital emergency report. Survival after 24 h was obtained by reviewing the medical reports of the respective patients. The annual admission rates were obtained from the public annual report of the Medical University.

#### 2.2.3. Variable Distribution

Statistical analyses were performed using SPSS (Version 22.0. Released 2013, Armonk, NY, USA: IBM Corp.). Data were checked for normal distribution using the Shapiro–Wilk test. Based on the data normality and the type of variables, results are presented as mean (standard deviation), frequency (percent), and median (interquartile range). Patients with a significant amount of missing data in their records were excluded.

### 2.3. Qualitative Methods

The qualitative part consisted of guided interviews with experts in in-hospital emergency medicine. In Austria, acute care hospitals are classified based on the scope of services they provide. Currently, there are
(1)Ninety-three standard hospitals providing basic healthcare services;(2)Twenty-eight specialized hospitals providing an extended range of medical services beyond basic care;(3)Seven central hospitals offering comprehensive care 24/7 for acute and complex medical needs.

#### 2.3.1. Selection of Participants

The qualification criteria for experts were defined as having practical experience of at least three years, occupying a leading function for at least five years or having published in the field of in-hospital emergency medicine before. For our study, we used the term “medical center” for specialized and central hospitals. Moreover, it was stipulated that these experts were employed in either of these hospitals.

Recruitment started using personal contacts from one author (BT) to the 7 central hospitals and the 28 specialized hospitals in Austria. These persons were requested to name members of their teams performing in-hospital emergency medicine regardless of their profession within one month. This revealed a sample of 41 persons. Using sector analysis, we screened to see if our predefined qualification criteria were met. Thereafter, 29 experts received an interview request, resulting in 5 interviews. This was followed by the snowball principle (i.e., asking respondents about other possible interview partners), leading to another five interviews [25,26].

#### 2.3.2. Collection of Qualitative Data

All authors defined the research objectives and identified the key issues. Thereafter, one of the authors (BT) developed an interview guideline using open-ended questions. The questionnaire draft underwent pretesting to confirm reliability and validity and to reduce the risk of subjective bias in the questionnaire. Finally, nine main and three follow-up questions addressed topics such as the structure of in-hospital emergency medicine, EWS, the mode of alerting, barriers when transferring EWS to Austria, the role of the management, and changes due to the pandemic. Finally, basic sociodemographics were categorized to ensure anonymity (see supplements for interview guide).

The guided interviews were conducted by one author (BT). After giving written consent, the location of the interview was determined at the request of the research participants, either using online video meetings (Zoom Communications, Inc., San Jose, CA, USA, 2023. Link: https://us05web.zoom.us/j/3464772323?pwd=cjVkTURmR2s2Q1VZSnUrSi9yRTd3dz09 (accessed on 15 April 2023, 13 June 2023)) or face-to-face meetings at the participants’ work places. During the interviews, alongside structured expertise, emphasis was placed on practical and procedural knowledge. Thus, this group of people possesses specialized ‘abstract’ expertise and exclusive knowledge in this subject area [25]. Furthermore, the interviewer contributed as a co-expert to discuss challenges at a high professional level. All interviews were audiotaped, transcribed, and subsequently deleted from the recording device.

Recruitment sought to include experts from each federal state as a minimum requirement to obtain meaningful statements. Moreover, a maximum of twenty interviews was sought to be required to reach theoretical saturation, given recent data. For better comparability, pediatricians were omitted as their patients are even more heterogeneous than adults.

#### 2.3.3. Qualitative Analysis

Initially, a first version was created with the help of Amberscript (Amberscript Global B.V., Amsterdam, The Netherlands), achieving a transcription accuracy of about 75–80%. All transcripts were checked multiple times and corrected for content accuracy. Additionally, linguistic smoothing was applied to statements influenced by dialect, as the primary interest was in the substantive content. Thereafter, a structured content analysis according to Mayring using the MAXQDA software (VERBI–Software, Consult Sozialforschung GmbH, Berlin, Germany) occurred [27]. First, in this systematic, rule-based qualitative method, a coding guideline was deductively derived from relevant literature in the field. Each category was defined and supplemented with coding rules. Thereafter, the textual content was segmented into coding units and assigned to the appropriate categories.

Thereafter, these categories were aligned with the four main components of in-hospital emergency systems (efferent and afferent limb, administrative part, quality management). Additional categories were derived from the interview guide. The analysis was conducted chronologically, following the course of the conversation. After coding the first half of the interviews, a review of the codes was conducted. Codes that were never applied were removed, and the descriptions were further refined. Finally, the emergent themes and main results were derived.

### 2.4. Integration of Quantitative and Qualitative Data

We chose the contiguous approach of integration through narrative and present the quantitative and qualitative findings side-by-side [28]. After reporting both types of data, an explanatory text links both datasets, thus allowing for a comprehensive interpretation. After identification of barriers to the implementation of EWS, we sought to provide a best practice tutorial, as many efforts to implement evidence-based innovations fail, even with highly developed implementation plans [29]. To avoid this, frameworks describing determinants such as barriers or facilitators are useful to develop much-needed context measures [30]. For this analysis, the Consolidated Framework for Implementation Research (CFIR) was used as it aims to predict or explain barriers and facilitators for implementation effectiveness [31]. Based on the five domains of the CFIR, we analyzed the results of the empirical data and existing literature [29]. As part of the last domain, the implementation process domain, we sought to provide practical guidance for the implementation of NEWS2 at Austrian medical centers. The generated insights from the interviews were then assigned to Kotter’s “eight steps for leading change” [23].

### 2.5. Ethics

The prospective part of the study was approved by the Research Committee for Scientific Ethical Questions (RCSEQ) of the Private University for Health Sciences and Health Technology, 6060 Hall in Tirol, and the Health University of Applied Sciences Tyrol, 6020 Innsbruck, Austria (protocol of 1 March 2023). All interviewed experts gave written consent before the interviews.

The retrospective part was approved by the Ethics Committee of the Medical University of Innsbruck, Austria (1151/2021). Patient consent was waived due to the retrospective study nature.

## 3. Results

### 3.1. Local Data Regarding IHCA in Innsbruck

From 1 January 2017 to 31 December 2022, 439,701 patients were admitted to an in-patient department at the medical center Innsbruck. In total, 27.7% of all verified in-hospital emergencies experienced an IHCA, 53.7% of all resuscitated patients experienced ROSC, and 32.8% survived for 24 h (see Figure 1).

The annual rates of in-hospital emergencies, IHCA, ROSC, and survival after 24 h are presented in Appendix A. Figure 2 presents the annual incidence of IHCA per 1000 in-patient admissions.

### 3.2. Data from the Guided Interviews

Ten experts from Austrian medical centers were interviewed, with an overall duration of 210 min and 21 s. As of the ninth interview, the provided answers revealed a high share of recurrence, thus showing signs of theoretical saturation [32].

Eight interviews were conducted using Zoom; two interviews were performed at the participants’ work places. For characteristics of the interviewed experts see Table 1.

The qualitative results are presented in chronological order.

### 3.3. Structure of In-Hospital Emergency Medicine in Austria

Most in-hospital emergency medicine in Austrian medical centers is performed by anesthesiologists and internists.


*“We are operating within that magnitude. Approximately half and half: anesthesiologists and internists.” (interview 1, line 279, consultant lead for in-hospital emergency medicine, board-certified specialist in anesthesiology and intensive care medicine, specialized hospital)*


Due to decentralized units, hospitals may have several CCO teams. Almost all hospitals use a standardized alerting mode (mostly a uniform phone number). Nearly one half use structured alerting criteria.


*“We have alert criteria, if you want to call them that. These were taken from Hillman’s work around 2000 or 2001. They’ve been tested since 2010 and have been in use since 2012. They’re what you call ‘single track and trigger’ systems.” (interview 4, line 61ff, critical care nurse, in-hospital emergency management coordination unit, central hospital)*


Interestingly, data showed a nearly complete absence of the usage of EWS in Austria, except of single track-and-trigger systems. Moreover, consistent quality management in in-hospital emergency medicine remains scarce.

### 3.4. Barriers to the Implementation of EWS

We identified the strained human resources after the pandemic, the fear of an increased workload for the CCO team and the lack of robust data regarding a benefit of survival as being the primary barriers.


*“I think it mainly fails because of nursing staff, simply because there isn’t enough capacity to check on every patient on the general ward once an hour or so. They’re just, probably everywhere, really at their limit.” (interview 9, line 238ff, board-certified specialist in anesthesiology and intensive care medicine, clinical responsibilities in in-hospital emergency medicine, central hospital)*


Moreover, with reorganization of the board-certified national nursing training, as of 2024 more care assistants were attending patients on in-patient wards. As the clinical assessment remains pivotal in the timely recognition of critically ill patients, less educated staff could lead to later recognition of such patients.


*“Another important aspect is the academization of nursing which was certainly not a beneficial development. Experience and clinical intuition remain essential to recognize a deteriorating patient.” (interview 8, line 160ff, head of department of anesthesiology, board-certified specialist in anesthesiology and intensive care medicine, specialized hospital)*


Finally, all experts perceived easy access to regular training as a facilitator for obtaining high quality of in-hospital emergency medicine.


*“It is important—indeed absolutely essential—that hospital staff possess certain knowledge and skill, especially in an acute care hospital. I believe that for quality assurance reasons, there is no way around maintaining a low-threshold access, and also offering regular training sessions.” (interview 4, line 225ff)*


In summary, the interviews provided data on the reasons for refraining from EWS and problems and challenges faced during implementation. For a successful implementation of EWS, numerous framework conditions in the complexity of acute care hospitals need to be considered. In the following paragraph we propose a solution as a “best practice tutorial”, which can be used as a guideline for implementing EWS on a local or a national basis.

### 3.5. Best Practice Tutorial

Based on our empirical data and the existing literature, we describe the five domains of the CFIR:Innovation domain

The first domain describes the early identification of critically ill patients before deterioration to reduce mortality. As neither patients nor CCO teams can predict when a patient’s condition will deteriorate, EWS should be implemented for all patients, especially on general wards. However, our survey revealed a missing distribution of EWS in Austria.


*“We do not use either MEWS or NEWS. They were temporarily used in one unit, but that happened to be an IMCU ward—essentially an environment for which these scores were neither intended nor designed. Consequently, they are no longer used there.”(interview 2, line 111ff, board-certified specialist in anesthesiology and intensive care medicine, clinical responsibilities in in-hospital emergency medicine, central hospital)*


Moreover, there is no nationwide plan to introduce them. With regard to the procedures for in-hospital emergencies, EWS contribute to a more efficient use of scarce resources and can minimize the costs of intensive care. Even if competencies are defined and theoretically available, clinical experience shows other problems with availability, motivation, and/or pronounced hierarchies. EWS can attenuate interprofessional collaboration.


*“Nursing staff is glad to have a structured concept in place, so they can say: you need to take care of this patient now because I have documented this value.” (interview 6, line 190ff, physician-in-charge of ICU, board-certified specialist in anesthesiology and intensive care medicine, responsible for the in-hospital emergency team, central hospital)*


Young colleagues in particular have well-founded reasons for requesting further clinical expertise for critically ill patients based on a score. This holds especially true given the aging population and the expected increase in the number of chronically ill patients, especially older ones [17].


*“Yes, but even the ward physician can be overwhelmed, because nowadays the spectrum of illnesses that patients bring with them is simply much broader.” (interview 10, line 277f, critical care nurse, intermediate care unit, department of orthopedic and trauma surgery, central hospital)*


Moreover, EWSs are particularly helpful in the case of pronounced hierarchical barriers [33].
2.Outer Setting domain

The outer setting domain describes the surroundings in which the inner setting exists. Factors from multiple outer settings have been identified:Process Management

Structured process management offers the opportunity to solve common problems (e.g., waiting times, missing medical reports) through clearly defined processes [18]. The administrative part of process management deals with the control and the provision of resources (e.g., material, personnel, budget) and a comprehensive training concept. The implementation of EWS initially incurs costs, and this is a frequently cited reason for not introducing EWS.


*“If people don’t exceed their maximum working hours, they can sign up for as many trainings as they want. That counts as working time, these are the costs.” (interview 5, line 3119ff, head of the emergency department, board-certified specialist in internal medicine, specialized hospital)*


However, higher staff qualification is associated with a lower mortality rate and can increase patient safety [34].
b.Quality Management

An increase in quality and productivity can only be achieved through strategic investments in staff training, digital infrastructure, and patient-centered care models. Furthermore, developing the quality of care is a top priority nationwide, uniformly across federal states, sectors, and professions. This needs robust data that have to be collected by (multi-)national registries like the German Resuscitation Registry.


*“We are members of the German Resuscitation Registry. We document our emergencies using a protocol provided by the German Resuscitation Registry. Two of my senior physicians consistently enter the data into the database. We thoroughly review the annual report and implement appropriate measures based on its findings.” (interview 7, line 149ff, head of department of anesthesiology, board-certified specialist in anesthesiology and intensive care medicine, specialized hospital)*


In the future, Austrian medical centers will have to prevent a reduction in the quality of in-hospital emergency care due to a shortage of staff by the implementation of an internationally validated EWS.
c.Hospital Management

Commitment of the management is pivotal in the introduction of an EWS.


*“Based on my impression, there would be a willingness to invest a certain amount of money if, an early warning system were to be implemented. This requires training, which incurs costs primarily due to the time spent on duty. I do believe such a willingness exists.” (interview 3, line 200ff, consultant lead for in-hospital emergency medicine, board-certified specialist in anesthesiology and intensive care medicine, specialized hospital)*


The primary goal of management must be to deliver high-quality emergency care to all patients and employees, utilizing available resources effectively. It also requires a willingness to invest considerable sums in the necessary staff training to ensure patient safety.
d.Society

From the point of view of society and health economics, as many patients as possible should be protected from serious consequential disabilities after in-hospital emergencies to maintain their workforce. Given the lack of further insight into in-hospital emergencies, a survey at federal level is warranted to obtain robust data.


*“And if such a requirement is centrally mandated by the ministry, it makes implementation significantly easier.” (interview 6, line 292f)*



3.Inner Setting domain


The inner setting domain describes the setting in which the innovation is implemented. Our data revealed that the current shortage of staff and the associated shortage of hospital beds may result in premature transfers to general wards, even though monitoring would still be necessary. The inconsistent electronic documentation across Austria was also identified as a barrier.


*“As a responder, one must have rapid and clear access to all relevant information. I require a well-organized overview of the patient’s condition over the preceding thirty minutes, which, of course, further supports the adoption of electronic solutions.” (interview 7, line 209ff)*


Therefore, we recommend a nationwide introduction of electronic documentation and the implementation of EWS on general wards. Given the available data we recommend NEWS2 as EWS.
4.Individual domain

This domain describes the roles and characteristics of individuals who are part of the implementation process of EWS:a.Emergency patients

All emergency patients are entitled to receive emergency care following best practice.
b.Alerting staff

The alerting staff requires expeditious and competent support. Additionally, thresholds for activating a CCO team should remain low to ensure timely intervention.


*“I would suggest lowering the threshold—that is, encouraging the staff to call whenever they have a concerning gut feeling.” (interview 1, line 221ff)*


As a higher level of staff qualification is associated with a lower mortality rate, access to regular training is recommended. In addition to training, clinical experience is also a key factor.


*“[…] What’s still essential is experience and clinical intuition—being able to tell if someone is unwell or has a problem. You only learn that by spending a lot of time directly with patients.” (interview 8, line 160ff)*



c.Emergency Teams [Deployed staff]


The CCO teams are mainly staffed by critical care personnel. Our results expressed the fear that the introduction of EWS could lead to a sharp increase in the number of alerts.


*“What should be avoided is a sudden increase in unjustified or non-indicated alerts. This would likely lead responders to start questioning the system as a whole.” (interview 3, line 178ff)*


Those who are confronted with emergency patients first and often alone should not be given a feeling of ignorance. Providing them a feeling of safety and certainty about how to proceed is the only way to secure an alert in time for the next emergency.
5.Implementation process domain

As seen in previous domains, multiple factors must be considered before implementing NEWS2. By assigning our findings to one of Kotter’s “eight steps for leading change” [23], we developed a structured guideline (see Figure 3).

## 4. Discussion

In this study, we first identified a surprisingly low incidence of in-hospital cardiac arrests (around 0.9/1000 in-patient hospital admissions) at our central hospital, which is nearly half of the reported rate across Europe (1.5–2.8/1000). This may be partly explained by the 24/7 availability of two highly specialized adult CCO teams with short response times, as such teams have been shown to decrease the rate of IHCA and, to a lesser extent, mortality. Clearly, the incidence of IHCA may be an inaccurate indicator of the quality of care, as IHCA is still rare, and fair comparisons need an adjustment for the case mix of patients [35]. Our observed ROSC rate of 54% is comparable with recent data. In addition, this meta-analysis reported 36% survival at 24 h, resembling our data. However, a high heterogeneity of the included studies hinders comparability, and knowledge of local data proves to be pivotal.

Interestingly, the introduction of NEWS in English hospitals was associated with a reduction in the rate of IHCA, but not mortality [10]. From a health economics perspective, evaluating the potential cost savings associated with the prevention of IHCA is reasonable. US registry data estimated first-year follow-up costs of USD 19,000 per patient after surviving an IHCA [36]. Such numbers support the reasoning towards management. Unfortunately, no such data is available for Austria.

Despite the observed low incidence, we feel that still too many patients face possibly preventable harm from IHCA. An underlying reason may be the still low usage of EWS in Austria as compared to the high level of EWS distribution in England [35]. Thus, the implementation of EWS is the next step needed in Austria. Prior to this, the reasons for the inconsistent reductions in incidence of IHCA or mortality after the implementation of EWS have to be addressed [13,14]. As part of the outer setting, barriers such as staff shortages, workload concerns, and lack of robust survival data were reported. This is in line with European workforce challenges and previous EWS implementation studies [16].

By 2050, the proportion of the very elderly population aged 85 and over in Austria will more than double. Thus, an aging population will lead to sicker and older patients, which is one reason for increased demands in national health systems [17].

Clearly, age alone should not be considered a risk factor for IHCA but should be combined with clinical status, comorbidities, and hospital-level factors (e.g., staffing, response systems) [37]. In addition, age was excluded in the development of NEWS, as its addition only marginally improved the area under the receiver operating characteristic curve [38]. Despite the historically high number of healthcare providers across Europe, an efficient deployment of personnel is needed [16].

As 80% of patients on general wards show a significant clinical deterioration in the 24 h before an IHCA, the usage of structured alert criteria or EWS allows for easier identification by staff on-site [3,5,39,40]. Clearly, this depends on a reliable assessment of vital signs by experienced nurses. However, our results revealing difficulties in expeditious detection of critical patients outside of intensive care units are in line with current literature [41]. Moreover, from the perspective of the individuals in the CFIR approach, a high level of regular training has been described as a facilitator by our experts.

From the perspective of the innovation domain, a question arises: what is the most appropriate EWS? Here we chose one of the best evaluated EWSs, the second version of the NEWS assessment tool [7], as it was developed in the UK, another European country. However, the notable differences between the UK health system and the Austrian system and others globally have to be kept in mind. Given the complexity of EWS, it is essential to pay attention to the weakness of NEWS2: it [a] requires trained staff, [b] consumes time, and [c] is susceptible to calculation errors [19,38]. However, considering these pitfalls ensures success by avoiding early missteps. This aligns with our findings of high-quality staff training and elaborate electronic documentation in hospitals as essential prerequisites. The use of an electronic EWS can reduce the workload and minimize the demonstrated fear of an increased workload [10]. Hogan and coworkers demonstrated that the conversion from paper to electronic NEWS usage was associated with a 7.6% decrease in IHCA rates in England [still with unchanged mortality rates] [35]. Moreover, this is in line with the appraisal of our experts. In summary, we agree with Hogan et al. concluding that the exact structure of the efferent response arm of EWS may not be as pivotal as a timely response.

Another barrier from the inner setting of the CFIR framework could constitute the fact that good performance of an EWS in some subgroups assumes this in all other clinical settings [15]. However, the latter is no evidence for the ineffectiveness of EWS; its correct application poses its true challenge. Possible risks may include the wrong prioritization of scores over clinical intuition with subsequent late responses in some cases or a lack of sensitivity in particular patient subgroups [42,43]. Our data, showing an inhomogeneous organization of in-hospital emergency medicine in Austria, hinders comparability already at a national level. Moreover, the Austrian context differs from countries with widespread EWS adoption (e.g., UK), highlighting the need for adaptation to national structures.

As the COVID-19 pandemic exacerbated pre-existing challenges like aging populations, increases in chronic diseases, increasing expectations from patients, etc., feasible solutions are needed [16]. Our findings reinforce previous evidence that using wearable devices for continuous monitoring of vital signs could ease a widespread implementation of EWS [21]. Moreover, web-based learning for early recognition improved assessment and management of deteriorating patients [44]. Such tools may further facilitate the introduction of an EWS. However, the impact of these new technologies upon incidence and mortality of IHCA has still to be investigated. Moreover, structured data collection at a federal level is warranted. This enables tailoring international EWS to national needs for improving survival after IHCA.

To ease the transfer of our results into clinical practice, we provide a practical guideline on the setting and circumstances of the implementation of an EWS. Our results demonstrated the apparent urgency for change and the nationwide implementation of EWS. Considering the main barriers and facilitators can create quick wins like a quicker assessment of vital signs [45]. Thereafter, ongoing data acquisition regarding IHCA and EWS performance is essential for the change, and aiming for continuous improvement remains essential.

### Limitations

When interpreting our results, several limitations have to be kept in mind. Clearly, our data being from only one European country, as well as the small number of qualitative interviews, limits the representativeness of our results. However, the theoretical saturation led to the cessation of recruitment after ten experts. We used personal contacts and snowballing for recruitment, which introduces the risk of selection bias. As only staff from Austrian medical centers were interviewed, this could bias our results and hinder comparability with other hospitals and healthcare systems. However, we sought to provide data that were as comparable as possible. Moreover, the development of the interview guide by one author can be seen as a limitation. Considering the voluntary and self-administered nature of the questionnaire, we do not dispute the possibility of potential response and self-selection bias in the open-ended questions of the survey. Moreover, particular interests of the respondents, e.g., a high pressure to cut costs, could have led to a subjective bias in our qualitative results.

Recruitment was rather tedious, which may have led to a disproportionate share of senior staff members as these were contacted. Moreover, as only one female expert was interviewed our data cannot be generalized to both sexes.

Finally, we cannot rule out the potential effect of missing data. We may have missed IHCA cases, as CPR in emergency rooms, critical care suites, etc., may have took place without alerting a CCO team. Furthermore, the rate of “do not resuscitate” orders may have changed. In summary, we emphasize that our findings are highly context-specific and not directly generalizable beyond Austrian hospitals.

## 5. Conclusions

We sought to investigate the reasons for inconsistent results on the effectiveness of EWS in the reduction in the rate of IHCA or mortality. Interviewing ten Austrian experts demonstrated an inhomogeneous organization of in-hospital emergency medicine and a low use of EWS in Austria. Moreover, strained staff resources after the pandemic, the fear of an increased workload, and the lack of reliable data on survival benefits were reported as the main barriers to the implementation of EWS. Therefore, adaptation to local settings is pivotal for a successful implementation of an established EWS. For structuring this complex process, we used the Consolidated Framework for Implementation Research to predict barriers and facilitators to implementation effectiveness. Proper staff training and elaborate electronic documentation are crucial facilitators, echoing evidence from English hospitals transitioning to electronic NEWS [10].

Application of our practical guideline should ease the implementation of NEWS2 in Austrian specialized and central hospitals. Ideally, this could reduce not only the incidence of IHCA but also mortality in this context. Such hospitals are advised to rethink their rapid response systems. The exact structure of the efferent response arm of EWS may be less important than the timely response on general wards ensured by an EWS.

As the recent pandemic fueled long-standing problems in healthcare, like an aging workforce, the time has come for effective measures. Regarding in-hospital emergency medicine, we urge Austrian politicians and stakeholders to introduce NEWS2 nationwide. The scientific community is needed to establish monitoring with a national registry for IHCA. Finally, future research should investigate the impact of new technologies on the assessment of early warning systems.

## Figures and Tables

**Figure 1 healthcare-13-02624-f001:**
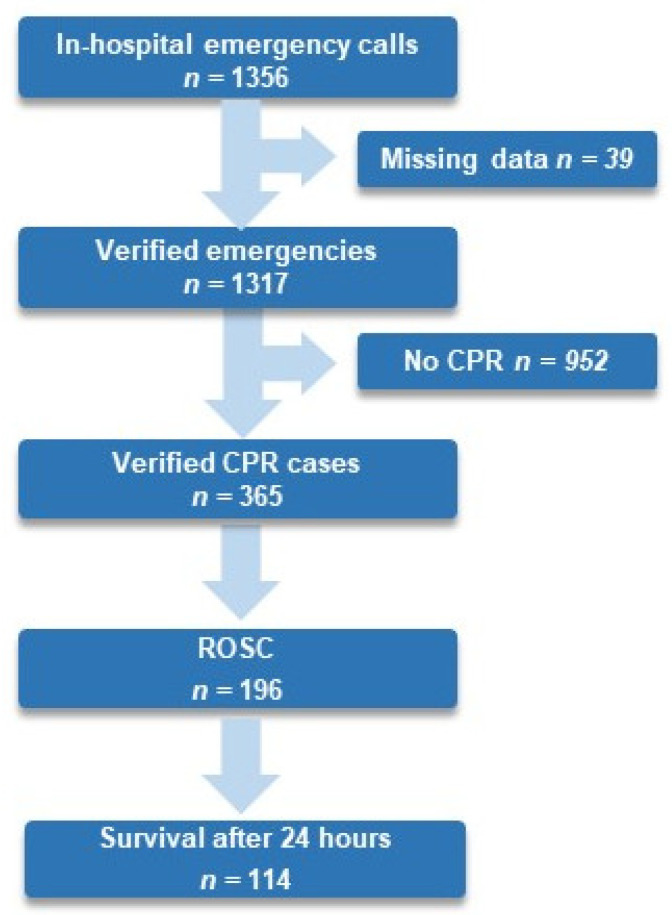
Flow chart of in-hospital emergencies, ROSC, and survival until 24 h after in-hospital cardiac arrest at Medical University Innsbruck from 1 January 2017 to 31 December 2022. Abbreviation: CPR, cardiopulmonary resuscitation; ROSC, return of spontaneous circulation.

**Figure 2 healthcare-13-02624-f002:**
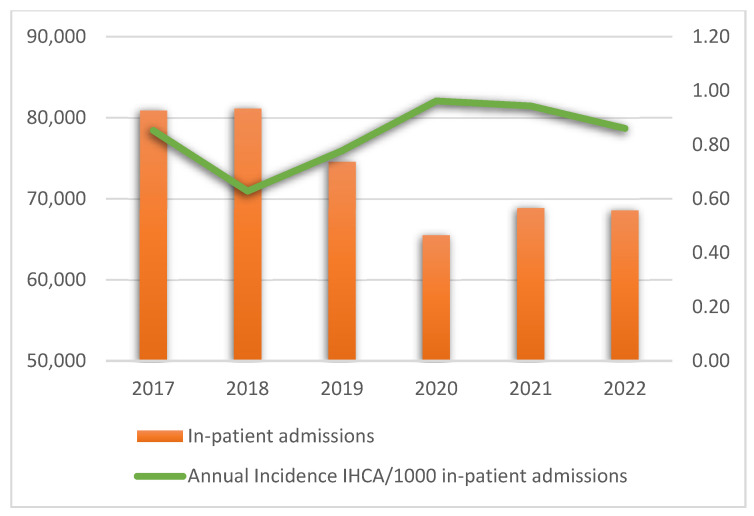
Annual incidence of in-hospital cardiac arrest per 1000 in-patient admissions. Abbreviation: IHCA, in-hospital cardiac arrest.

**Figure 3 healthcare-13-02624-f003:**
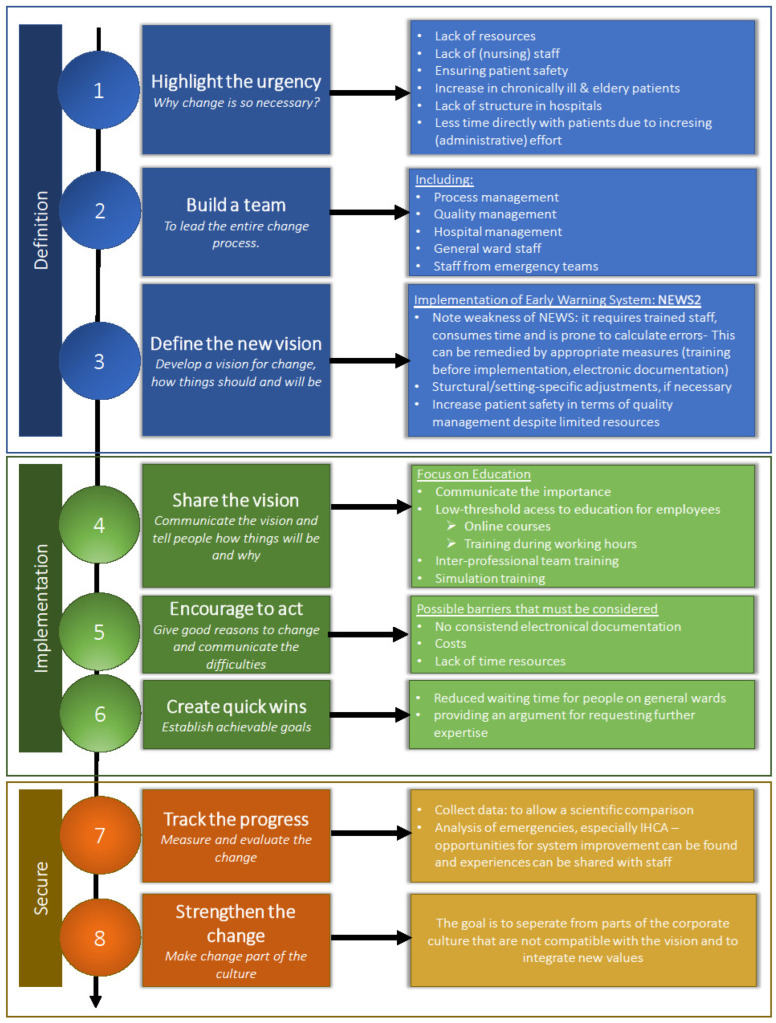
Practical guideline for implementation of NEWS at Austrian medical centers based upon the “eight steps for leading change” (according to Kotter) [23].

**Table 1 healthcare-13-02624-t001:** Characteristics of the interviewed experts [n = 10].

Characteristic		Number of Participants
Sex	Female	1
	Male	9
Age	31–40 years	2
	41–50 years	5
	51–60 years	2
	>60 years	1
Working experience	≤3 years	0
	4–7 years	0
	8–15 years	2
	>16 years	8
Experience in in-hospital emergency medicine	3–5 years	1
	6–8 years	0
	9–11 years	3
	>12 years	6
Profession	Nurse practitioner	2
	Doctor	8
Hospital beds at occupation of the participant	>500 beds	4
	>1000 beds	6

## Data Availability

The datasets used and analyzed during the current study can be made available from the corresponding author on reasonable request due to institutional policies.

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
