# Peer review of "Overcoming Barriers in the Introduction of Early Warning Scores for Prevention of In-Hospital Cardiac Arrests in Austrian Medical Centers†"

_healthcare, 2025, doi:10.3390/healthcare13202624_

Round 1

Reviewer 1 Report (Previous Reviewer 2)

Comments and Suggestions for Authors

I would like to begin by congratulating the authors for their consistent effort and dedication throughout the revision and improvement of this manuscript. The current version demonstrates a solid level of academic rigor and coherence, which significantly enhances its value. I commend the authors for producing a remarkable manuscript that makes an important contribution to the field.

Author Response

Reviewer 2 Report (New Reviewer)

Comments and Suggestions for Authors

I thank the authors for the opportunity to read this interesting paper. Analyzing why EWS are not used in hospital settings is very interesting. However, this study has a major limitation: it only refers to a specific country, Austria, which may have local conditions that may not be applicable to other settings. The recruitment of respondents is highly questionable as it begins with people who may have ties of friendship and interests with the author. Furthermore, the selection criteria for respondents are highly questionable: only people with practical or managerial experience (3-5 years) or people who have published an article in this field of hospital emergencies are admitted. A person, for example, a collaborating researcher or statistical expert, may appear as the author of an article without knowing anything about EWS. I am also concerned that the authors indicate that they are experts in hospital emergencies and survey people who may not work in hospitals, but in "medical centers," which the authors differentiate from hospitals. Of the 29 experts selected, only 5 initially participated. It is unknown whether the participants were from the same hospital or hospitals in the same area (this should be clarified). In the end, only 10 interviews were obtained. Can 10 interviews with people who were not selected randomly or by stratification reflect the state of the art in Austria? I think not. The authors indicate that 8 experts were the minimum requirement to obtain meaningful statements. How did they calculate this? Do they ensure that the 8 experts cover all hospitals in Austria? I am surprised that the exclusion criteria for analyzing the incidence of AHCI were that the patient did not speak German or English or was not Austrian. What could have motivated the authors to include this exclusion criterion? Weren't these patients (regardless of origin or language) who had an AHCI in the hospital? I don't think it is an acceptable exclusion criterion for this part of the study, and this criterion should be eliminated and the calculations reformulated. Could it be that the authors have a low incidence of AHCI due to the exclusion of these cases? On the other hand, the authors talk about survival but don't define it. 24-72 hours, 7 days, one month, or hospital discharge? That needs to be defined. We don't know if the discussion and conclusions are a sample of what happens in Austria or in a specific region or city, since we don't know how many hospitals are represented. Furthermore, the feelings of a hospital professional don't represent what might happen there, since there may be particular interests involved, especially when the sample was selected in such a targeted manner (friendship or prior acquaintance).

Author Response

Reviewer 3 Report (New Reviewer)

Comments and Suggestions for Authors

The proposed study ‘Overcoming barriers in the introduction of early warning scores for prevention of in-hospital cardiac arrests in Austrian medical centers – a mixed methods approach’ presents an interview-based analysis of the Austrian hospital context with reference to a specific patients’ condition, in-hospital cardiac arrest, and its management by personnel.

There are significant flows in the manuscript, such as:

1) The introduction could benefit of some elaboration. In particular, the concept of EWS should be better described in its general formulation (what data it usually relies on, what are the common alert levels, when is it usually implemented in the patient management workflow etc.); also, the specific study aim should be more clearly outlined.

2) Methods described in 2.2.3 should be renamed to ‘Variables distribution’, as, according to the description, defining it ‘statistical analysis’ is somehow far-stretched.

3) Paragraph 2.3.3: it is not totally clear what authors mean by ‘data analysis’. They refer to some categories, which however are not listed. Please elaborate this section to better clarify what processing was applied to collected interview data.

4) A description of the performed interview is missing from the text.

5) Line 130 ‘resulting in five interviews’ is in contrast with line 226 ‘Ten experts from Austrian medical centers were interviewed’. Please clarify.

6) I wasn’t able to find the table 2 referenced in line 230.

7) Results section is based on reporting some quotes from the interviews, whose structure is not disclosed (see comment 4). This appears to be somewhat of an arbitrary selection: it is totally unclear how these quotes were selected, why they are relevant, and how this analysis serves the study aim (which is itself unclear, see comment 1). In particular, it is not clear how the study led to the definition of the proposed guidelines reported in figure 3. Moreover, the characteristics of the suggested EWS are not defined.

8) The description of the NEWS and NEWS2 should be moved from the discussion to the introduction.

9) The Discussion section should clarify whether the study successfully addressed its research aim, explain how the findings support or contradict the original objectives, and discuss the underlying reasons for these outcomes in the context of existing knowledge and of the expected impact of the research (eventually, also in term of real-world applications). This part is missing from the manuscript.

10) The conclusions are not clearly supported by the results (and their interpretation through discussion).

11) Check typos (e.g. line 55)

Round 2

Reviewer 2 Report (New Reviewer)

Comments and Suggestions for Authors I thank the authors for the effort they have made in answering the questions raised by the reviewers and for implementing the changes in the paper that I believe have significantly improved it.

Author Response

Thank you again for your valuable comments. These have enhanced our work, and we are now delighted that our results have been accepted for publication! 

Reviewer 3 Report (New Reviewer)

Comments and Suggestions for Authors

Authors have addressed all the raised issues.

Comments on the Quality of English Language

Please carefully check for typos and grammar inconsistencies, which are still present in the text.

Author Response

Thank you again for your valuable comments. These have enhanced our work, and we are now delighted that our results have been accepted for publication. 

We have also carefully checked the text for spelling and grammatical errors and corrected the inconsistencies.

This manuscript is a resubmission of an earlier submission. The following is a list of the peer review reports and author responses from that submission.

Round 1

Reviewer 1 Report

Comments and Suggestions for Authors

The content of the abstract needs to be organized concisely. Additionally, the conclusion is written in a casual tone.

  • Please revise the introduction to clearly outline the research background, the necessity of the study, and the research objectives. Points i) and ii) may be omitted.
  • The method does not need to mention 'As part of a master thesis (BT)'.

Please review the guidelines for completion. Additionally, the numerical brackets used to denote references differ from the notation style of this journal.

  1. Introduction: It is essential to accurately supplement the content. Additionally, it is important to include information regarding the current status of existing research, as well as the limitations and barriers associated with the introduction process of EWS.
  2. Materials and Methods: Describe your research methodology in accordance with the guidelines for reporting mixed methods, referencing previous studies. 2.1 Study Design 2.2 Quantitative Methods 2.2.1 Study Subjects and Sample Size 2.2.2 Instruments, 2.2.3. Data collection methods, 2.2.4. Statistical analysis methods, 2.3. Qualitative methods. 2.3.1. Selection of Participants, 2.3.2. Data Collection Method, 2.3.3. Methodology of data analysis, etc. 2.4. Please describe how you integrated quantitative and qualitative data.
  3. Results: The figure should be numbered below the figure. Tables 1 and Figure 1 are duplicated. The results for Figure 2 are insufficient. Typos should be corrected, and the results should be clearly stated in the study methodology, including that all interviews were conducted by the authors. The 'Best practice tutorial' section explains the theoretical framework (CFIR) that the researchers used to analyze the data and the rationale for selecting that framework. This is a crucial aspect of clarifying the methodological approach of the study. The reference to "analysis of empirical data and results from existing literature" pertains to the analysis process and is more appropriately placed in the methods section. The research results section should present specific findings derived from data analysis. It is essential to demonstrate the outcomes of the analysis within the CFIR framework, rather than merely stating that CFIR was utilized. ### Reason: Improved clarity, corrected grammatical errors, and enhanced vocabulary for better readability and technical accuracy.

The 8-step visualization model presented in the study is quite appealing. However, since the process of developing this model was conducted using scientific writing methods, there are several omissions in the text that need to be addressed. Additionally, it is important to make an effort to complete the submission form with careful consideration of its requirements.

Author Response

Referee #1:
The content of the abstract needs to be organized concisely. Additionally, the conclusion is written in a casual tone.

In response: The abstract has been rewritten. Moreover, we rephrased the conclusion to be more concise (page 1).

Please revise the introduction to clearly outline the research background, the necessity of the study, and the research objectives. Points i) and ii) may be omitted.

In response: We now state the objectives of the study more clearly. The points i) and ii) have been omitted as they provide no additional information (page 2).

The method does not need to mention 'As part of a master thesis (BT)'.

In response: This has been deleted in the abstract as well as in the methods section (page 1, abstract and page 3, methods).

Please review the guidelines for completion. Additionally, the numerical brackets used to denote references differ from the notation style of this journal.

In response: Thank you for these advices. We have changed the reference of the master thesis and now denote all references with square brackets.

Introduction: It is essential to accurately supplement the content. Additionally, it is important to include information regarding the current status of existing research, as well as the limitations and barriers associated with the introduction process of EWS.

In response: We included more information on current research and now cite 3 more studies:

  • Dall’Ora et al. 2021. How Long Do Nursing Staff Take to Measure and Record Patients’ Vital Signs Observations in Hospital? A Time-and-Motion Study. doi:10.1016/j.ijnurstu.2021.103921.
  • Briggs et al. 2024. Safer and More Efficient Vital Signs Monitoring Protocols to Identify the Deteriorating Patients in the General Hospital Ward: An Observational Study. doi:10.3310/HYTR4612.
  • Van Velthoven et al. 2023. ChroniSense National Early Warning Score Study: Comparison Study of a Wearable Wrist Device to Measure Vital Signs in Patients Who Are Hospitalized. doi:10.2196/40226.

Moreover, we provide information about barriers when introducing an EWS (page 2, introduction).

Materials and Methods: Describe your research methodology in accordance with the guidelines for reporting mixed methods, referencing previous studies. 2.1 Study Design 2.2 Quantitative Methods 2.2.1 Study Subjects and Sample Size 2.2.2 Instruments, 2.2.3. Data collection methods, 2.2.4. Statistical analysis methods, 2.3. Qualitative methods. 2.3.1. Selection of Participants, 2.3.2. Data Collection Method, 2.3.3. Methodology of data analysis, etc. 2.4. Please describe how you integrated quantitative and qualitative data.

In response: The reviewer suggests a more thorough description of our research methodology. We have completely rewritten the methods section to provide a better picture of our study design (see pages 3 - 5).

Results: The figure should be numbered below the figure.

In response: The figure captions have been moved below all figures, thank you.

Tables 1 and Figure 1 are duplicated.

In response: We agree that the information in table 1 and figure 1 overlap to some extent. Our intention was to show the course of the annual rates of admissions, in-hospital emergencies, IHCA and survival of IHCA in table 2. After discussing it with all authors and re-thinking the suggestion from the reviewer we chose to move table 1 into supplements (now Supplemental table S2).

The results for Figure 2 are insufficient.

In response: As the annual rates of admissions decreased distinctly (~20%) from 2018 to 2020 we aimed to show the course of the incidence of IHCA. Could you specify what results in figure 2 are insufficient. Maybe the display of 2 different units in the axis?

Typos should be corrected, and the results should be clearly stated in the study methodology, including that all interviews were conducted by the authors.

In response: We had our manuscript again proofread and corrected all typos found. We have included the information that all interview were conducted by one author (page 3, methods section, heading 2.3.2; page 6, results section, heading 3.2).

The 'Best practice tutorial' section explains the theoretical framework (CFIR) that the researchers used to analyze the data and the rationale for selecting that framework. This is a crucial aspect of clarifying the methodological approach of the study. The reference to "analysis of empirical data and results from existing literature" pertains to the analysis process and is more appropriately placed in the methods section.

In response: Thank you for this suggestion, this part has been moved to the methods section (see page 4, heading 2.4).

The research results section should present specific findings derived from data analysis. It is essential to demonstrate the outcomes of the analysis within the CFIR framework, rather than merely stating that CFIR was utilized. ### Reason: Improved clarity, corrected grammatical errors, and enhanced vocabulary for better readability and technical accuracy.

In response: The reviewer points out a somewhat vague presentation of the qualitative results. Therefore, we have now included numerous citations of the conducted interviews in all parts of the qualitative results sections. This includes the chapter with the CFIR as well (see page 6-10, heading 3.3 – 3.5).

The 8-step visualization model presented in the study is quite appealing. However, since the process of developing this model was conducted using scientific writing methods, there are several omissions in the text that need to be addressed.

In response: Thank you for this suggestion. We have included the description of creating the guideline in the methods section (page 4, heading 2.4) and changed the corresponding results part (page 10, last para).

Additionally, it is important to make an effort to complete the submission form with careful consideration of its requirements.

In response: We carefully reviewed the submission form and strove to meet all requirements. 

All authors wish to thank you for reviewing our work!

Reviewer 2 Report

Comments and Suggestions for Authors

First of all, I would like to begin by congratulating the authors for the work they have put into writing this manuscript. The chosen topic is of great interest, both in terms of highlighting the need to introduce this tool/score (EWS/NEWS) for identifying potentially critical patients into clinical practice, and in terms of the difficulties encountered—such as the reluctance, I would say—in implementing this score in everyday clinical work.

I recommend the authors consider the following:

1.     One of the strengths of this manuscript is the very subject it addresses. The EWS score is recommended for identifying patients at risk of critical deterioration and is also used to determine the frequency of monitoring and the intervention of rapid response teams. In our daily pursuit of the "new," we often forget that we already have simple tools at hand that can ease medical decision-making and, at the same time, increase the safety of care provided to our patients.

2.     This score is primarily aimed at medical personnel in departments that care for patients with diverse pathologies and who may not be familiar with interventions for critically ill patients with unstable vital functions, as intensive care units and emergency departments are. The goal is, of course, a noble one—to identify early those patients who may deteriorate and to ultimately reduce morbidity and mortality rates. Here, the persistent work of rapid response teams comes into play, as they must convince both management and healthcare staff of the benefits and utility of implementing this score into clinical practice.

3.     Another strength of this manuscript lies in the way the authors addressed this critical issue identified within their hospital and in the fact that they sought higher-level expertise—namely, expert opinion. Identifying weaknesses and providing solutions to improve them adds significant value to the research presented.

4.     Furthermore, I am convinced that this issue is not limited to AUSTRIA alone but is relevant in many countries, regardless of their geographic location. It is difficult to change decades of practice and to implement a new tool. To be more explicit, think of Peter Safar, who introduced BLS (Basic Life Support) more than 50 years ago in hospitals and to the general public. Today's healthcare services are still advocating for this concept and emphasizing its importance in survival after cardiac arrest (OHCA and IHCA).

5.     As a weakness of this manuscript, I would mention the small number of experts - only 10, the fact that it is a single-center study, and the lack of a broader dissemination of the opinion survey to the medical staff of various hospital departments, except for intensive care units (where the pressure lies and some reluctance to implement a new score might have been expected).

6.     In line 163, you mention the Zoom platform as the method used for conducting online interviews. Please specify the application link, the year of launch/version, the country of origin/headquarters address, and the date the link was accessed.

7.    Given the nature of the study and the use of questionnaires, I would suggest, in the section on the limitations of the study, additional clarification regarding selection bias. One possible example would be: ,, onsidering the voluntary and self-administered nature of the questionnaire, we do not dispute the possibility of potential response and self-selection bias in the open-ended questions of the survey,,.

Author Response

Referee #2:

Comments and Suggestions for Authors

First of all, I would like to begin by congratulating the authors for the work they have put into writing this manuscript. The chosen topic is of great interest, both in terms of highlighting the need to introduce this tool/score (EWS/NEWS) for identifying potentially critical patients into clinical practice, and in terms of the difficulties encountered—such as the reluctance, I would say—in implementing this score in everyday clinical work.

I recommend the authors consider the following:

  1. One of the strengths of this manuscript is the very subject it addresses. The EWS score is recommended for identifying patients at risk of critical deterioration and is also used to determine the frequency of monitoring and the intervention of rapid response teams. In our daily pursuit of the "new," we often forget that we already have simple tools at hand that can ease medical decision-making and, at the same time, increase the safety of care provided to our patients.

  1. This score is primarily aimed at medical personnel in departments that care for patients with diverse pathologies and who may not be familiar with interventions for critically ill patients with unstable vital functions, as intensive care units and emergency departments are. The goal is, of course, a noble one—to identify early those patients who may deteriorate and to ultimately reduce morbidity and mortality rates. Here, the persistent work of rapid response teams comes into play, as they must convince both management and healthcare staff of the benefits and utility of implementing this score into clinical practice.

  1. Another strength of this manuscript lies in the way the authors addressed this critical issue identified within their hospital and in the fact that they sought higher-level expertise—namely, expert opinion. Identifying weaknesses and providing solutions to improve them adds significant value to the research presented.

In response: Thank you for your congratulations and your in-depth analysis! Our aim was to get a thorough picture of in-hospital cardiac arrests before implementing an EWS to avoid barriers and pitfalls.

  1. Furthermore, I am convinced that this issue is not limited to AUSTRIA alone but is relevant in many countries, regardless of their geographic location. It is difficult to change decades of practice and to implement a new tool. To be more explicit, think of Peter Safar, who introduced BLS (Basic Life Support) more than 50 years ago in hospitals and to the general public. Today's healthcare services are still advocating for this concept and emphasizing its importance in survival after cardiac arrest (OHCA and IHCA).

In response: Oh yes, you are right, enhancing patient safety is a long and difficult road.

  1. As a weakness of this manuscript, I would mention the small number of experts - only 10, the fact that it is a single-center study, and the lack of a broader dissemination of the opinion survey to the medical staff of various hospital departments, except for intensive care units (where the pressure lies and some reluctance to implement a new score might have been expected).

In response: When planning this study we felt 20 interviews to be feasible. Interestingly, we observed theoretical saturation after the ninth interview and therefore stopped recruitment. Clearly this is a small sample size. Moreover, we included only medical centers from one country to get data being as comparable as possible. As both issues may be seen as a limitation this has been added to the limitations section (page 13, 4th paragraph).

  1.    In line 163, you mention the Zoom platform as the method used for conducting online interviews. Please specify the application link, the year of launch/version, the country of origin/headquarters address, and the date the link was accessed.

In response: We have included the suggested information in the corresponding section (page 4, heading 2.3.2).

  1.   Given the nature of the study and the use of questionnaires, I would suggest, in the section on the limitations of the study, additional clarification regarding selection bias. One possible example would be: ,, onsidering the voluntary and self-administered nature of the questionnaire, we do not dispute the possibility of potential response and self-selection bias in the open-ended questions of the survey.

In response: The reviewer provides another possible limitation. Thank you for the fine wording, this has been included (page 13, 4th paragraph).

All authors wish to thank you for reviewing our work!